# Association between Dietary Fat Intake and Hyperuricemia in Men with Chronic Kidney Disease

**DOI:** 10.3390/nu14132637

**Published:** 2022-06-25

**Authors:** Fumika Oku, Akinori Hara, Hiromasa Tsujiguchi, Keita Suzuki, Kim-Oanh Pham, Fumihiko Suzuki, Sakae Miyagi, Masaharu Nakamura, Chie Takazawa, Kuniko Sato, Toru Yanagisawa, Takayuki Kannon, Atsushi Tajima, Hiroyuki Nakamura

**Affiliations:** 1Department of Hygiene and Public Health, Graduate School of Medical Science, Kanazawa University, 13-1 Takaramachi, Kanazawa 920-8640, Japan; okufumika@yahoo.co.jp (F.O.); ahara@m-kanazawa.jp (A.H.); t-hiromasa@med.kanazawa-u.ac.jp (H.T.); takazawa@staff.kanazawa-u.ac.jp (C.T.); sato3kuniko@yahoo.co.jp (K.S.); yanagisa1break@yahoo.co.jp (T.Y.); 2Department of Hygiene and Public Health, Faculty of Medicine, Institute of Medical, Pharmaceutical and Health Sciences, Kanazawa University, Kanazawa 920-8640, Japan; keitasuzuk@gmail.com (K.S.); kimoanhpham129@gmail.com (K.-O.P.); f-suzuki@den.ohu-u.ac.jp (F.S.); m.nakamura.83-7-7@r.vodafone.ne.jp (M.N.); 3Advanced Preventive Medical Sciences Research Center, Kanazawa University, 1-13 Takaramachi, Kanazawa 920-8640, Japan; kannon@med.kanazawa-u.ac.jp (T.K.); atajima@med.kanazawa-u.ac.jp (A.T.); 4Community Medicine Support Dentistry, Ohu University Hospital, Koriyama 963-8611, Japan; 5Innovative Clinical Research Center, Kanazawa University, 13-1 Takaramachi, Kanazawa 920-8641, Japan; smiyagi@staff.kanazawa-u.ac.jp; 6Department of Bioinformatics and Genomics, Graduate School of Advanced Preventive Medical Sciences, Kanazawa University, 13-1 Takaramachi, Kanazawa 920-8640, Japan

**Keywords:** chronic kidney disease, hyperuricemia, dietary fats, fatty acids, the brief-type self-administered diet history questionnaire

## Abstract

Despite a close relationship between chronic kidney disease (CKD) and uric acid level, few studies have examined the relationship between uric acid level and fat intake by kidney function status. Therefore, we investigated the association between dietary fat intake and hyperuricemia with and without decreased kidney function in males living in Shika Town, Ishikawa Prefecture, Japan. This study included 361 males with a mean age of 60.7 years. Dietary fat and fatty acid intakes were evaluated using the brief-type self-administered diet history questionnaire. Reduced kidney function was defined as an estimated glomerular filtration rate (eGFR) <60 mL/min/1.73 m^2^, while hyperuricemia was defined as a serum uric acid level >7.0 mg/dL. A two-way analysis of covariance showed that saturated fatty acid (*p* = 0.026), monounsaturated fatty acid (*p* = 0.014), and polyunsaturated fatty acid (*p* = 0.022) were significantly lower in the high uric acid group than in the normal uric acid group. In multiple logistic analysis stratified by renal function, lipid intake was negatively associated with hyperuricemia in the low eGFR group. These findings suggest that higher dietary lipid/fatty acid intake may be effective in the prevention and treatment of hyperuricemia in men with CKD.

## 1. Introduction

Hyperuricemia is a common biochemical abnormality observed in 20–25% of adult males and only a few females [1,2]. The prevalence of gout induced by hyperuricemia has been reported to be as high as 7% in males aged of ≥65 years [3]. In addition to gout, the pathogenesis of hyperuricemia has been implicated in visceral fat accumulation and insulin resistance, both of which are closely related to organ damage such as atherosclerosis, cardiovascular disease, and chronic kidney disease (CKD) [4].

Dietary factors that contribute to elevated serum uric acid levels and the development of gout include increased intake of sugar [5], alcohol [6,7], and purines [8]. Therefore, lifestyle modification is essential for hyperuricemia with or without drug therapy. To optimize energy intake, measures such as restriction of alcohol consumption and avoidance of excess purine intake have been recommended in addition to dietary therapy [9,10]. Regarding dietary style, the Dietary Approaches to Stop Hypertension (DASH) diet and the Mediterranean diet reduce serum uric acid [11,12,13]. However, only a few studies have investigated the relationship between serum uric acid levels and dietary lipids [14,15,16]. An intervention study in healthy young adults revealed that n-3 polyunsaturated fatty acid intake with 2 g/day of fish oil for 8 weeks reduced serum uric acid levels [14]. Another intervention study in healthy older males reported that a 3-month intake of 700 mg/day of omega-3 lipid supplements reduced serum uric acid levels [15]. Regarding dietary style, a 16-week intervention with a calorie-restricted diet in which saturated fatty acids were substituted for monounsaturated and polyunsaturated fatty acids was shown to induce weight loss and reduce serum uric acid levels in males with gout [16]. However, all of these intervention studies used dietary supplements or diets with adjusted dietary lipid composition, and none examined the direct relationship between dietary lipid intake and serum uric acid levels.

Given the importance of dietary therapy for hyperuricemia according to kidney function from the viewpoint that the kidneys play a key role in regulating serum uric acid levels [17], this study aimed to clarify the relationship between dietary lipids and hyperuricemia stratified by kidney function in males from a community-based cross-sectional study.

## 2. Materials and Methods

### 2.1. Study Population

The study participants were residents of Shika Town, Ishikawa, Japan, who underwent a medical checkup between April 2013 and March 2018. This cross-sectional study was conducted as part of the Shika study [18,19,20]. The participants were middle-aged and older adults aged of ≥40 years living in two model districts (Horimatsu and Higashimasuho).

A total of 1184 residents received medical checkups during the study period. Of these, 1175 underwent uric acid measurement, among whom 1161 did not receive treatment for hyperuricemia between 2011 and 2012, the years preceding the start of the medical checkups. Among the 1161 individuals, 808 responded to the brief-type self-administered diet history questionnaire (BDHQ) [21]. Nutrient intake was converted to a percentage of daily energy intake using the density method. Seven hundred and ninety-nine participants had a daily caloric intake of >600 kcal and <4000 kcal. Additionally, uric acid levels were classified into two groups according to reference values. As only 8 of the females were in the high uric acid group, 361 males were included in the final analysis (Figure 1).

### 2.2. Demographic Data and Medical History

Information on the demographic characteristics of the participants, including age, sex, family composition, job status, educational background, exercise habit, and smoking habit, was collected using a questionnaire [18]. Family composition was classified according to living alone or not. Educational background was classified into the following four levels: 1, elementary and junior high school; 2, high school; 3, junior college or vocational school; and 4, university and higher. Exercise habits were categorized on a 5-point scale based on the weekly frequency of exercise: 1, daily; 2, 5–6 days per week; 3, 3–4 days per week; 4, 1–2 days per week; and 5, none. Smoking habits were classified according to whether the individuals was a current smoker or not. Additionally, participants were asked questions relating to their diabetes and hypertension treatment histories [18].

### 2.3. Assessment of Kidney Function and Hyperuricemia

To evaluate kidney function, the estimated glomerular filtration rate (eGFR) was calculated using the following formula [22]:eGFR (mL/min/1.73 m^2^) = 194 × serum creatinine level^−1.094^ × age^−0.287^ (× 0.739 for females)

An eGFR of <60 mL/min/1.73 m^2^ was defined as reduced kidney function, based on the definition of the Clinical Practice Guideline for Evaluation and Management Chronic Kidney Disease of KDIGO 2012 [23]. Serum creatinine concentrations were measured using an enzymatic method.

The definition of hyperuricemia was based on the “Guidelines for the Treatment of Hyperuricemia and Gout” by the Japanese Society for Gout and Uric & Nucleic Acids [10]. Regardless of sex, participants with uric acid levels of >7.0 mg/dL measured by the enzyme (uricase) method were defined as the high uric acid group, while those with uric acid levels <7.0 mg/dL were defined as the normal uric acid group.

### 2.4. Evaluation of Nutrient Intake

The BDHQ was used to evaluate nutrient intake [21]. The BDHQ is a brief version of the diet history questionnaire, asking study participants about their consumption frequency of 58 food and beverage items. These food and beverage items were selected from foods commonly consumed in Japan, mainly from a food list used in the National Health and Nutrition Survey of Japan. A previous study in a Japanese population compared the estimated energy and nutrient intake calculated using the BDHQ with data obtained from a 16-day diet record. The energy-adjusted intake of the 42 nutrients were correlated with the diet record, and the Pearson correlation coefficients were 0.45 to 0.61 in females, and 0.41 to 0.63 in males.

The BDHQ has been validated in previous studies as a suitable method for assessing nutrient intake in the Japanese population [24,25]. This study used fat intake adjusted for estimated energy intake. The fat intake consisted of total fat (% energy), animal fat (% energy), plant fat (% energy), saturated fatty acid (SFA) (% energy), monounsaturated fatty acid (MUFA) (% energy), polyunsaturated fatty acid (PUFA) (% energy), n-3 PUFA (% energy), and n-6 PUFA (% energy). Alcohol intake was also assessed from the BDHQ.

### 2.5. Other Variables

Participants’ body mass index (BMI) was calculated using the following equation:BMI = weight (kg)/[height (m)]^2^

### 2.6. Statistical Analysis

The Student’s *t*-test was performed to examine the relationships between continuous variables, while the chi-square test was used to investigate the relationships between categorical variables. A two-way analysis of covariance (ANCOVA) was conducted to examine the main effects and interactions between uric acid level and eGFR value on fat and fatty acid intake. The following covariates were used in this study regarding the epidemiological survey by Choi et al. [26]: age, job status, educational background, exercise habit, current smoker, alcohol intake, BMI, diabetes treatment, and hypertension treatment. Furthermore, multiple logistic regression analysis was performed, stratifying the two groups into normal kidney function and reduced kidney function groups, with the two uric acid groups as the dependent variable and fat and fatty acid intake as the independent variable, adjusted for the same covariates as in the two-way ANCOVA.

The *p*-values shown in all analyses were two-tailed, with a *p*-value < 0.05 being considered statistically significant. IBM SPSS^®^ Statistics 26.0 (SPSS Inc., Armonk, NY, USA) was used to perform statistical analysis.

## 3. Results

### 3.1. Participant Characteristics

Table 1 shows the characteristics of the 361 analyzed participants. The mean age (SD) of the participants was 60.7 (10.0) years. Age (*p* < 0.001), BMI (*p* = 0.029), uric acid level (*p* = 0.004), and the proportion of hypertension treatment (*p* = 0.015) were significantly higher in the reduced kidney function group than those in the normal kidney function group. However, the proportion of current smokers (*p* = 0.010) and alcohol intake (*p* = 0.005) were significantly lower in the reduced kidney function group compared to the normal kidney function group. Among fat intake, SFA intake in the reduced renal function group was significantly higher than that in the normal kidney function group (*p* = 0.013).

### 3.2. Comparison between the Two Uric Acid Groups

Table 2 shows a comparison between the normal uric acid group and the high uric acid group. Alcohol intake (*p* = 0.009) and BMI (*p* = 0.048) were significantly higher in the high uric acid group than in the normal uric acid group. In contrast, age (*p* = 0.020), the proportion of diabetes treatment (*p* < 0.001), and eGFR (*p* = 0.014) were significantly lower in the high uric acid group than in the normal uric acid group. Among fat intake, plant fat intake (*p* = 0.025) was significantly lower in the high uric acid group than in the normal uric acid group.

### 3.3. Analysis of Covariance

Table 3 shows the results of the two-way ANCOVA for the two uric acid and two kidney function groups on fat and fatty acid intake. Covariates were adjusted for age, job status, education background, exercise habits, current smoker, alcohol intake, BMI, diabetes treatment, and hypertension treatment. There was an interaction between serum uric acid levels and kidney function on dietary fat intake. Specifically, in the reduced kidney function group, with the exception of animal fat and n-3 fatty acids, all fat and fatty acid intakes in the high uric acid group were significantly lower than those in the normal uric acid group, whereas, in the normal kidney function group, this relationship was not found between the two uric acid groups.

### 3.4. Association between Uric Acid Level and Fat Intake after Stratification into Two eGFR Groups

Table 4 shows the multiple logistic regression analysis of the relationship between fat and fatty acid intake and high uric acid levels, stratified by normal and reduced kidney function groups. Covariates were adjusted for the same variables as in the two-way ANCOVA. With the exception of animal fat, fat and fatty acid intake were inversely related to hyperuricemia in the reduced kidney function group, whereas no such relationship was observed in the normal kidney function group.

## 4. Discussion

This study examined the association between dietary fat and fatty acid intake and hyperuricemia stratified by kidney function in middle-aged and older Japanese males. Our findings revealed that fat and fatty acid intake, including SFA, MUFA, and PUFA, was inversely related to hyperuricemia in males with reduced kidney function. Therefore, higher fat and fatty acid intake is likely to lower serum uric acid levels in males with reduced kidney function.

The high uric acid group had higher alcohol intake, higher BMI, and lower eGFR than the normal uric acid group. Alcohol consumption, obesity, and CKD are well-known risk factors for hyperuricemia and gout [4,9,27]. Additionally, plant fat in the high uric acid group was significantly lower than that in the normal uric acid group. Similarly, PUFA intake was lower in the high uric acid group. Previous studies examining the relationship between fatty acid intake and serum uric acid levels include a randomized controlled trial comparing the effects of fish oil intake containing n-3 PUFA [14], an intervention study using n-3 PUFA supplements [15], and an intervention study using a calorie-restricted diet in which SFA was replaced with unsaturated fatty acids [16]. The difference in the fatty acid intake between the two uric acid level groups observed in this study seems to be consistent with these previous studies.

Previous reports on dietary style restricting the consumption of fats and meats include the DASH diet and Mediterranean diets, which are well-known to be effective for achieving appropriate uric acid levels, and have different uric acid-lowering effects, depending on patient characteristics [28,29,30]. A randomized, crossover feeding trial by Juraschek et al. [28] demonstrated that the DASH diet lowered serum uric acid among participants with a higher uric acid level at baseline. Chatzipavlou et al. [29] performed an interventional study using a Mediterranean diet in patients with asymptomatic hyperuricemia, and the findings showed that a higher baseline uric acid level was correlated with a greater reduction in uric acid level after the intervention. Furthermore, a recent cross-sectional study by Gao et al. [30] reported that the DASH diet was effective in lowering uric acid levels in a subgroup with the following characteristics: aged ≥ 50 years, male, and low physical activity. Thus, since the effect of a diet adjusted for fat intake on uric acid levels is inconstant, we speculate that fat intake may be more effective for individuals who are prone to high uric acid levels.

Our notable finding was an inverse relationship between fat and fatty acid intake and a high serum uric acid level in participants with reduced kidney function, whereas these relationships were not observed in the normal kidney function group. Juraschek et al. [28] reported that in African American hypertensive adults, the higher the baseline uric acid level, the greater the uric acid-lowering effect of the DASH diet. It is speculated that the population with high uric acid levels included many adults who were susceptible to hyperuricemia, such as those with gout and CKD. In the Juraschek et al. study [28], the baseline patient characteristics included a mean BMI of 34.7 kg/m^2^, a prevalence of diabetes of 29.1%, and a mean number of antihypertensive medications of 1.8 on the medication regimen, which suggests a high-risk population for CKD. In this study, the reduced kidney function group had a mean uric acid level of 6.3 mg/dL, which was significantly higher than the 5.8 mg/dL recorded for the normal kidney function group; the higher fat and fatty acid intake is presumed to be more effective in the reduced kidney function group. These findings suggest that higher fat and fatty acid intake for hyperuricemia is effective in individuals with reduced kidney function, suggesting the need for future consideration of individualized dietary therapy.

A mechanism linking fat intake and uric acid metabolism may involve the effects of dietary fats on insulin sensitivity. The effects of dietary fats on insulin sensitivity have been examined in metabolic syndrome with underlying insulin resistance [31,32,33,34]. A systematic review [31] of 14 observational studies and 16 clinical trials demonstrated that many of the observational studies found beneficial associations between MUFA/PUFA intake and metabolic syndrome components. Moreover, previous clinical trials support the benefits of MUFA- or PUFA-enriched diets in reducing metabolic syndrome. In addition to the abovementioned results, both the HOMA-R and blood insulin level, indices of insulin resistance, were shown to be reduced [31]. Another clinical trial [32] in 472 subjects with metabolic syndrome from eight European countries reported that the highest HOMA-R group, one of three groups classified by HOMA-R levels, had the greatest reduction in HOMA-R and insulin levels after consumption of a high MUFA diet or a diet supplemented with n-3 PUFA. These effects with a high MUFA diet or a diet supplemented with n-3 PUFA were also associated with the reduced plasma level of the inflammatory cytokine, IL-6 [32]. Insulin resistance disturbs glycolysis and uric acid clearance by the kidneys, leading to increased production of uric acid and decreased urinary uric acid clearance, respectively [33,34]. A case-control study of nonalcoholic fatty liver disease in Caucasians indicated that dietary patterns high in unsaturated fatty acids [35] were associated with lower serum uric acid levels and improved insulin resistance. A systematic review of insulin resistance in CKD by Spoto et al. [36] explains that the Janus effect of adipokines stimulates fatty acid oxidation by activating AMP-activated protein kinase, thereby improving insulin sensitivity. Therefore, we speculate that the mechanism in which fat and fatty acid intake is negatively associated with hyperuricemia in reduced kidney function partially involves the improvement of insulin resistance.

This study has several limitations that warrant discussion. First, as this is a cross-sectional study, additional longitudinal studies are needed to elucidate causal relationships. Second, the BDHQ may lack objectivity as it was self-administered. Third, as only males were evaluated, it is unclear whether similar results would be obtained for women.

## 5. Conclusions

The results of this study demonstrate that fat and fatty acid intake was inversely related to the higher uric acid level in males with reduced eGFR. This finding suggests that a higher intake of dietary fat and fatty acids is effective in the prevention and treatment of hyperuricemia in males with CKD.

## Figures and Tables

**Figure 1 nutrients-14-02637-f001:**
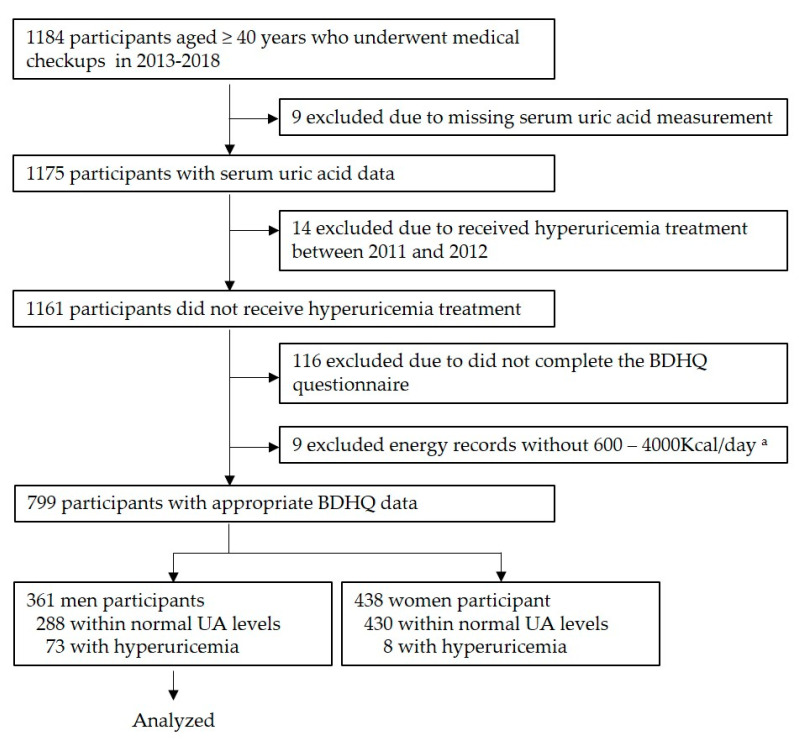
Participant recruitment chart. ^a^ This reference value was chosen for the following reasons: less than 600 kcal/day is equivalent to half the energy intake required for the lowest physical activity category; more than 4000 kcal/day is equivalent to 1.5 times the energy intake required for the medium physical activity category. Abbreviations: BDHQ, brief-type self-administered diet history questionnaire.

**Table 1 nutrients-14-02637-t001:** Participant characteristics.

	Total (*N* = 361)		Normal Kidney Function (*n* = 291)		Reduced Kidney Function (*n* = 70)		*p*-Value ^a^
	Mean/*n*	*SD*/%	Mean/*n*	*SD*/%	Mean/*n*	*SD*/%	
Age, years	60.73	9.96	59.45	9.65	66.04	9.53	<0.001
No occupation, *n*	89	24.65	65	22.34	24	34.29	0.058
Living alone, *n*	18	4.99	12	4.12	7	10.00	0.104
Education ^b^, *n*	2.10	1.03	2.14	1.02	1.94	1.03	0.152
Exercise ^c^, *n*	3.94	1.42	3.98	1.42	3.79	1.39	0.336
Current smoker, *n*	123	34.07	108	37.11	15	21.43	0.010
Alcohol intake, % energy	4.66	4.61	4.95	4.76	3.47	3.68	0.005
BMI, kg/m^2^	23.49	2.74	23.34	2.69	24.14	2.89	0.029
Diabetes treatment, *n*	29	8.03	23	7.90	6	8.57	0.854
Hypertension treatment, *n*	108	29.92	79	27.15	30	42.86	0.015
Uric acid, mg/dL	5.86	1.38	5.76	1.35	6.28	1.40	0.004
eGFR, mL/min/1.73 m^2^	70.75	14.83	75.65	11.06	50.41	10.60	0.000
Total fat, % energy	21.50	5.60	21.35	5.50	22.11	5.97	0.310
Animal fat, % energy	10.11	3.85	9.97	3.82	10.72	3.93	0.143
Vegetable fat, % energy	11.39	3.39	11.39	3.34	11.39	3.61	0.989
SFA, % energy	5.54	1.70	5.43	1.63	5.99	1.89	0.013
MUFA, % energy	7.60	2.19	7.56	2.16	7.76	2.31	0.502
PUFA, % energy	5.45	1.40	5.48	1.39	5.34	1.46	0.464
n-3 PUFA, % energy	1.20	0.43	1.20	0.43	1.20	0.44	0.904
n-6 PUFA, % energy	4.23	1.12	4.26	1.11	4.11	1.74	0.333

Abbreviations: SD, standard deviation, BMI, body mass index, SFA, saturated fatty acid, MUFA, monounsaturated, fatty acid, PUFA, polyunsaturated fatty acid. ^a^: *p*-values were calculated using the Student’s *t*-test and chi-squared test for continuous and categorical variables, respectively. ^b^: 1, elementary and junior high school; 2, high school; 3, junior college or vocational school; 4, university and higher. ^c^: 1, daily; 2, 5–6 days per week; 3, 3–4 days per week; 4, 1–2 days per week; 5, none.

**Table 2 nutrients-14-02637-t002:** Comparison between the normal uric acid group and the high uric acid group.

	Normal Uric Acid (*n* = 288)				High Uric Acid (*n* = 73)				*p*-Value ^a^
	Mean/*n*	SD/%	95%CI		Mean/*n*	SD/%	95%CI		
			Lower	Upper			Lower	Upper	
Age, years	61.34	10.08	60.18	62.50	58.32	9.14	56.22	60.41	0.020
No occupation, *n*	72	25.00	20.31	30.38	16	21.92	12.36	31.47	0.545
Living alone, *n*	12	4.17	1.85	6.48	6	8.22	1.87	14.56	0.243
Education ^b^, *n*	2.11	1.01	1.99	2.23	2.07	1.08	1.82	2.33	0.809
Exercise	3.94	1.42	3.78	4.11	3.94	1.41	3.61	4.27	0.998
Current smoker, *n*	92	31.94	26.13	36.99	31	42.47	31.18	54.53	0.089
Alcohol intake, % energy	4.34	4.42	3.83	4.85	5.92	5.10	4.75	7.09	0.009
BMI, kg/m^2^	23.35	2.76	23.03	23.67	24.06	2.60	23.46	24.66	0.048
Diabetes treatment, *n*	29	10.07	0.07	0.14	0.00	0.00	0.00	0.00	<0.001
Hypertension treatment, *n*	86	29.86	24.24	34.79	23	31.51	20.78	42.24	0.741
Uric acid, mg/dL	5.41	1.04	5.29	5.53	7.64	1.07	7.40	7.89	<0.001
eGFR, mL/min/1.73 m^2^	71.90	13.69	70.32	73.48	66.22	18.08	62.07	70.37	0.014
Total fat, % energy	21.75	5.54	21.11	22.39	20.54	5.76	19.22	21.86	0.099
Animal fat, % energy	10.16	3.95	9.70	10.61	9.94	3.47	9.15	10.74	0.669
Vegetable fat, % energy	11.59	3.35	11.20	11.98	10.60	3.45	9.80	11.39	0.025
SFA, % energy	5.60	1.70	5.41	5.80	5.26	1.67	4.88	5.64	0.123
MUFA, % energy	7.67	2.15	7.42	7.92	7.33	2.32	6.80	7.86	0.236
PUFA, % energy	5.52	1.39	5.36	5.68	5.18	1.40	4.86	5.51	0.071
n-3 PUFA, % energy	1.21	0.43	1.16	1.26	1.14	0.41	1.04	1.23	0.166
n-6 PUFA, % energy	4.28	1.12	4.15	4.41	4.03	1.12	3.77	4.29	0.086

Abbreviations: SD, standard deviation, CI, confidence interval, BMI, body mass index, SFA, saturated fatty acid, MUFA, monounsaturated, fatty acid, PUFA, polyunsaturated fatty acid. ^a^: *p*-values were calculated using the Student’s *t*-test and chi-squared test for continuous and categorical variables, respectively. ^b^: 1, elementary and junior high school; 2, high school; 3, junior college or vocational school; 4, university and higher.

**Table 3 nutrients-14-02637-t003:** Two-way ANCOVA for the two uric acid and two kidney function groups on fat and fatty acid intake.

		Normal Uric Acid (*n* = 288)			High Uric Acid (*n* = 73)			*p*-Value ^a^		
		Mean	95% CI		Mean	95% CI		*P*1	*P*2	*P*3
			Lower	Upper		Lower	Upper			
Total fat, % energy	NKF	21.38	20.72	22.04	21.46	19.95	22.98	0.017	0.438	0.011
	RKF	22.78	21.26	24.29	18.76	16.45	21.07			
Animal fat, % energy	NKF	10.00	9.52	10.49	10.24	9.13	11.35	0.183	0.627	0.081
	RKF	10.74	9.63	11.85	8.90	7.21	10.60			
Vegetable fat, % energy	NKF	11.37	10.97	11.77	11.23	10.31	12.14	0.020	0.489	0.039
	RKF	12.04	11.12	12.96	9.86	8.46	11.26			
SFA, % energy	NKF	5.46	5.27	5.66	5.47	5.02	5.92	0.031	0.781	0.026
	RKF	6.07	5.62	6.51	5.01	4.32	5.69			
MUFA, % energy	NKF	7.52	7.26	7.79	7.62	7.01	8.23	0.033	0.584	0.014
	RKF	8.14	7.53	8.74	6.64	5.71	7.56			
PUFA, % energy	NKF	5.47	5.30	5.65	5.46	5.07	5.86	0.021	0.056	0.022
	RKF	5.54	5.14	5.93	4.55	3.95	5.16			
n-3 PUFA, % energy	NKF	1.21	1.16	1.27	1.22	1.09	1.34	0.125	0.062	0.097
	RKF	1.19	1.07	1.32	0.98	0.79	1.17			
n-6 PUFA, % energy	NKF	4.24	4.10	4.38	4.22	3.91	4.53	0.021	0.098	0.026
	RKF	4.33	4.01	4.64	3.56	3.08	4.04			

Abbreviations: CI, confidence interval, BMI, body mass index, SFA, saturated fatty acid, MUFA, monounsaturated, fatty acid, PUFA, polyunsaturated fatty acid, NKF, normal kidney function, RKF, reduced kidney function. ^a^: A two-way analysis of covariance. Adjusted for age, occupation, living alone, education, exercise, current smoking, alcohol intake, BMI, diabetes treatment, hypertension treatment. *P*1: two uric acid groups, *P*2: two kidney function groups, *P*3: two uric acid and two kidney function groups.

**Table 4 nutrients-14-02637-t004:** Multiple logistic regression analysis of the relationship between fat and fatty acid intake and high uric acid levels.

		Odds Ratio	95% CI		*p*-Value ^a^
			Lower	Upper	
Normal kidney function (*n* = 291)	Total fat	1.00	0.93	1.07	0.910
	Animal fat	1.01	0.92	1.11	0.792
	Vegetable fat	0.97	0.87	1.09	0.616
	SFA	0.99	0.78	1.26	0.943
	MUFA	1.01	0.85	1.19	0.939
	PUFA	0.98	0.75	1.26	0.850
	n-3 PUFA	1.02	0.45	2.36	0.954
	n-6 PUFA	0.96	0.69	1.33	0.789
Reduced kidney function (*n* = 70)	Total fat	0.83	0.70	0.99	0.033
	Animal fat	0.83	0.67	1.04	0.107
	Vegetable fat	0.80	0.63	1.02	0.070
	SFA	0.54	0.31	0.96	0.035
	MUFA	0.56	0.33	0.94	0.030
	PUFA	0.45	0.22	0.94	0.033
	n-3 PUFA	0.15	0.02	1.33	0.089
	n-6 PUFA	0.39	0.16	0.96	0.041

Abbreviations: CI, confidence interval, SFA, saturated fatty acid, MUFA, monounsaturated, fatty acid, PUFA, polyunsaturated fatty acid. ^a^: Multiple logistic regression analysis. Adjusted for age, occupation, living alone, education, exercise, current smoking, alcohol intake, BMI, diabetes treatment, hypertension treatment.

## Data Availability

Data in the present study are available upon request from the corresponding author. Data are not publicly available due to privacy and ethical policies.

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
