# Peer review of "Association between Dietary Fat Intake and Hyperuricemia in Men with Chronic Kidney Disease"

_nutrients, 2022, doi:10.3390/nu14132637_

Round 1
Reviewer 1 Report
The manuscript is innovative and well written, the methods are clear and standardized. However, it is necessary to carry out small revisions:
1. Introduction
a) Line 66-78, it is very redundant to be an introduction, it should be reduced;
b) the purpose of the study should be better stated in the introduction
Author Response
Response to Reviewer 1 Comments
Comments and Suggestions for Authors
The manuscript is innovative and well written, the methods are clear and standardized. However, it is necessary to carry out small revisions:
Comment 1
- Introduction
- a) Line 66-78, it is very redundant to be an introduction, it should be reduced;
Response 1
We have amended the introduction section of the revised manuscript as follows: "Since serum uric acid levels are highly dependent on the relative balance of uric acid reabsorption and secretion in the proximal tubules of the kidney [17], hyperuricemia is likely to be present due to decreased ability to excrete uric acid clinically [18]. A recent meta-analysis investigating the association between hyperuricemia/gout and dietary factors highlighted the need for subgroup analysis by kidney function that interacts with hyperuricemia [19]. Thus, it is crucial to examine the relationship between hyperuricemia and diet, especially lipid and fatty acid intake, stratified by renal function, to consider renal function-based dietary therapy for hyperuricemia." (P2 L66-73)
Comment 2
- b) the purpose of the study should be better stated in the introduction
Response 2
We have amended the introduction section of the revised manuscript as follows: "This cross-sectional study investigated the relationship between dietary fat and fatty acid intake and hyperuricemia separately according to reduced kidney function in males from a community-based longitudinal observational study in residents in Shika Town (the Shika study)." (P2 L74-77)

Reviewer 2 Report
dear Authors
The article has an excellent design and your results are clearly supported. The limitation of BDHQ which may lack objectivity, shows that more studies are needed to confirm your conclusions.
This paper can be acccepted for publication, in present form.
Author Response
Response to Reviewer 2 Comments
Comments and Suggestions for Authors
dear Authors
Comment 1
The article has an excellent design and your results are clearly supported. The limitation of BDHQ which may lack objectivity, shows that more studies are needed to confirm your conclusions.
This paper can be accepted for publication, in present form.
Response 1
We appreciate your very dedicated review of our article.
